

# Offshore Crustal Thickness Variation along the Palu–Koro Strike–Slip Fault in the Sulawesi region from OBS Receiver Function Analysis

Tingwei Yang[1,2], ChuanChuan Lü[3], Tianyao Hao[1,2], Nicholas Rawlinson[3], Tao Xu[1], Sri Widiyantoro[4], Alfian Alfian[5], Muhammad Taufiq Rafie[6], David Prambudi Sahara[6]

[1] Institute of Geology and Geophysics, Chinese Academy of Sciences, Beijing, 100029, China
[2] University of Chinese Academy of Sciences, Beijing 100049, China
[3] Bullard Labs, Department of Earth Sciences, University of Cambridge, Cambridge CB3 0EZ, UK
[4] Global Geophysics Research Group, Faculty of Mining and Petroleum Engineering, Institut Teknologi Bandung, Bandung, Indonesia
[5] Seismology Exploration and Engineering Research Group, Faculty of Mining and Petroleum Engineering, Institut Teknologi Bandung, Bandung, Indonesia
[6] Geophysical Engineering Department, Faculty of Mining and Petroleum Engineering, Institut Teknologi Bandung, Bandung, Indonesia

*Correspondence to*: C. Lü (chuanchuanlu@esc.cam.ac.uk)

**Abstract.** The North Sulawesi Subduction Zone is one of the youngest active subduction systems in the western Pacific. In western Sulawesi, the Palu–Koro strike–slip fault connects with the westward-extending North Sulawesi Trench, forming a distinctive subduction–transform fault system. Understanding the crustal structure beneath the Celebes Sea and the geometry of the Palu–Koro fault is crucial for assessing regional deformation, rupture dynamics, and seismic hazards. In this study, we analyse data from nine ocean bottom seismometers (OBSs) deployed across the Palu–Koro fault using the receiver function H–κ stacking method to estimate crustal thickness. Our results reveal a shallow Moho (~8 km depth) beneath the Celebes Sea, in contrast to significantly greater depths (~25 km) beneath eastern Kalimantan and northern Sulawesi. Sharp variations in Moho depth near the Palu–Koro fault suggest the juxtaposition of two distinct crustal blocks. Combining S-wave velocity structures and local seismicity catalogue, we infer that the Palu–Koro fault is a left-lateral, supracrustal strike–slip fault extending into the Celebes Sea. These findings provide new geophysical constraints on the interplay between strike–slip faulting and subduction retreat, with implications for the generation of tsunamis by submarine earthquakes in this tectonically complex region.

## 1 Introduction

The Celebes Sea region lies at the convergence of three major tectonic plates: the Indo-Australian, Pacific, and Eurasian plates (Hamilton, 1979) (Fig. 1). This tectonic setting has produced a highly complex lithospheric structure, with the Palu–Koro fault acting as a key boundary controlling regional deformation. As a major left-lateral strike–slip fault, the Palu–Koro fault



accommodates the differential motion between the Makassar Block to the west and the North Sula Block to the east. These blocks exhibit opposing rotational patterns with respect to the Sunda Plate: the Makassar Block rotates counter-clockwise at

~1°/Ma, while the North Sula Block rotates clockwise at a faster rate of ~4°/Ma (Vigny et al., 2002). This rotational divergence produces a strong sinistral shear zone along the Palu–Koro fault. Over the past ~30 million years, the Palu–Koro fault has played a central role in crustal segmentation and block interactions in Sulawesi (Socquet et al., 2006; Spakman and Hall, 2010). It also forms a kinematic link between continental deformation in western Sulawesi and subduction at the North Sulawesi Trench. GPS observations show high present-day slip rates along the fault (30–46 mm/yr), confirming its significance in the

regional strain budget (Walpersdorf et al., 1998; Stevens et al., 1999). However, the offshore structure and geometry of the Palu–Koro fault—particularly in the Celebes Sea—remain poorly defined due to limited geophysical constraints.

Geodetic data, offset geomorphological features, and hydrothermal activity collectively demonstrate that the Palu–Koro fault is an actively deforming left-lateral strike–slip system (Hamilton, 1979; Walpersdorf et al., 1998; Cipta et al., 2021). On 28

September 2018, a Mw 7.5 earthquake nucleated along this fault and generated a tsunami that severely impacted western Sulawesi (Fox et al., 2021). Focal mechanism solutions indicate NNW-trending sinistral strike–slip events with a normal faulting component (Wang et al., 2019). Supershear rupture propagation was confirmed by both teleseismic waveform inversion and geodetic modelling, suggesting predominantly horizontal displacement (Bao et al., 2019; Song et al., 2019; Wu et al., 2021). Subsequent studies have refined the structural interpretation of the fault system. Song et al. (2019) identified

rupture segmentation, with an onshore left-lateral segment and an offshore NW–SE-striking fault exhibiting oblique-slip characteristics. Coulomb stress modelling by Liu et al. (2021) indicated enhanced static stress along adjacent fault segments, potentially contributing to aftershock activity and tsunami generation. Cui et al. (2021) interpreted the fault as occupying a regional strike–slip stress regime with a subordinate extensional component, favouring supershear rupture initiation. The offshore continuity of the fault and its structural linkage to the North Sulawesi subduction zone remain critical for

understanding rupture dynamics and associated tsunami hazards. The extension of the Palu–Koro fault offshore off-shore is likely to be one of the main causes of the tsunami produced by this earthquake. Therefore, it is vital for the science community to locate the extension of the Palu–Koro fault in the Celebes Sea and understand its characteristics to better assess the geohazard it poses to the region.

Given its tectonic importance and potential seismic hazard, characterising the offshore extent of the Palu–Koro fault and the crustal structure it offsets is essential. To address these, we utilise broadband ocean bottom seismometer (OBS) to record earthquakes offshore in the southern Celebes Sea and Makassar Strait to investigate the crustal structure, then analysis using receiver function method to estimate crustal thickness and the Vp/Vs ratios beneath nine OBS stations crossing the Palu–Koro fault. Additionally, we integrate local seismicity to delineate fault geometry and assess its role in upper crustal deformation in

the Celebes Sea region, thus we aim to provide new geophysical insights into the active tectonics of the Sulawesi region and the nature of fault–trench interaction at a complex plate boundary.



## 2 Data and Method

### 2.1 Earthquake data

A total of 27 OBSs, leased from the Institute of Geology and Geophysics, Chinese Academy of Sciences (IGGCAS), were
deployed across the study area from August 2019 to August 2020 as part of the collaboration project between the University
of Cambridge and the Institut Teknologi Bandung. The instruments were spaced approximately 50–70 km apart. However,
due to the premature ageing of the rubber valve on the pressurized glass cabin—attributed to changes in the supply chain—
many OBS units experienced slow leakage. Compounded by delays in the recovery voyage caused by COVID-19 restrictions,
only 12 OBSs were successfully recovered (Rawlinson et al., 2020). Of these, three OBSs have disk formatting issues affecting
the horizontal components, leaving only nine stations suitable for receiver function analysis (Fig. 1).

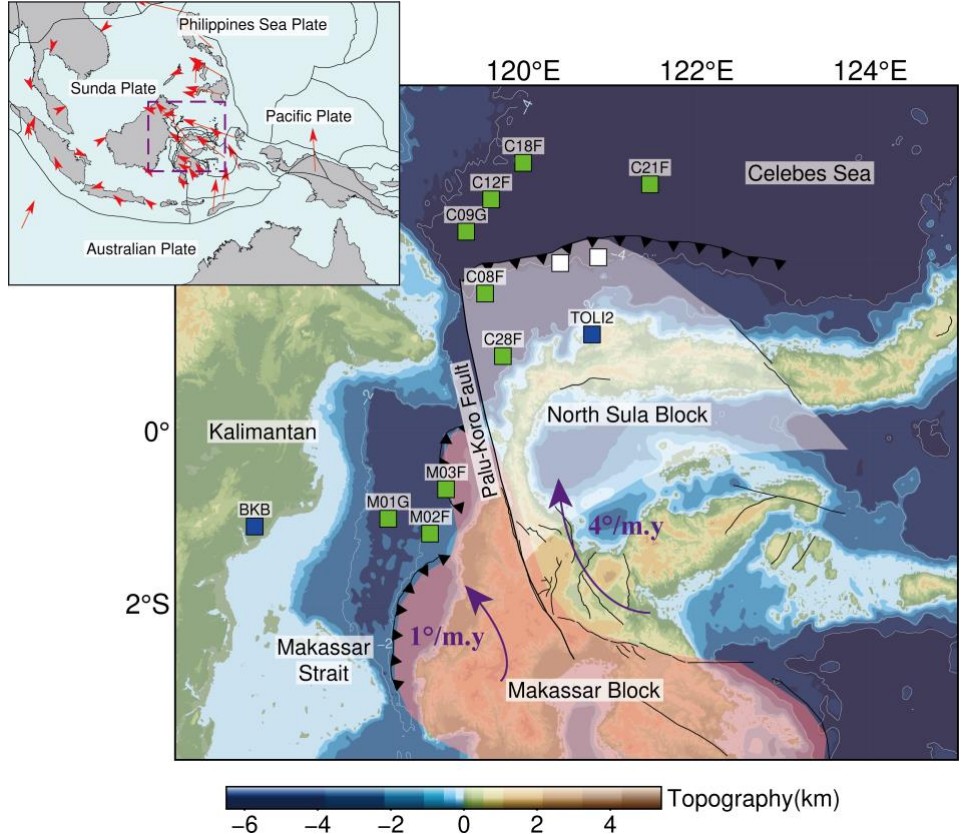

**Figure 1: Locations of OBSs and land stations in the Celebes Sea and Makassar Strait (The green square represent OBSs and the blue square is land station; The white and red pad represent three different geological microblocks as North Sula Block and Makassar Block; The purple arrow represents the direction of rotation of the block indicated by the GPS study; Figure in the upper**
**left corner indicate the study area in Southeast Asia; The red arrows represent the motion of the plates; The black solid line represent the plate boundaries).**



Unsurprisingly, the signal-to-noise ratio (SNR) of the OBS data is much lower than that of land stations, particularly in the low-frequency range. This SNR degradation is primarily attributed to poor coupling between the sensors and the seafloor, as

well as elevated ambient noise levels in the marine environment (Collins et al., 2001) (Fig. S4-S5). In addition, due to seasonal variations in temperature and pressure, along with strong surface currents during the deployment period, the OBS units are affected by two key issues: internal clock drift and uncertainty in instrument azimuth. Together with the low SNR, these factors have traditionally limited the utility of OBS datasets (Niu and Li, 2011; Wang et al., 2016; Le et al., 2018).

Accurate timing and reliable three-component seismic records are critical for a range of modern seismological applications. To correct for timing errors, we applied ambient noise cross-correlation to estimate and adjust for clock drift following Hable et al. (2018). The time correction result of one example is shown in Fig. S1. Azimuthal misalignment of the horizontal components was corrected by minimising energy on the transverse component of teleseismic P-waves (Niu and Li, 2011; Wang et al., 2016). An example of the horizontal orientation correction result is shown in Fig. S2. To validate these corrections,

we used a teleseismic event with a moment magnitude of 6.5 and an epicentral distance of 49.5° (Fig. 2b). Despite improvements, the quality of the teleseismic waveforms recorded by the OBS stations remains notably lower than that of a nearby land station, which displays significantly cleaner signal characteristics (Fig. 2c). After processing, we demonstrated the data characteristics using a time-frequency analysis approach (Fig. S3). Table 1 summarises the basic parameters and processing outcomes for the nine OBSs and two land-based stations used in this study.


**Table 1: OBS deployment parameters and timing errors (Hori represents the deviation of the azimuth of the East component; Time_Err represents the clock error during the entire observation period; Dt_Err represents the clock error for each sampling point.)**

| Station | Lat (°) | Lon (°) | Depth (m) | Hori (°) | Time_Err (s) | Dt_Err (s) |
|---------|---------|---------|-----------|----------|--------------|------------|
| C28F | 0.871 | 119.766 | -2400.7 | -73.2 | -7.5911 | -4.92E-09 |
| C12F | 2.689 | 119.631 | -4800.4 | 62.8 | -7.2212 | -4.66E-09 |
| C21F | 2.862 | 121.461 | -4500.4 | 23.9 | -2s.9025 | -1.86E-09 |
| C18F | 3.112 | 120.001 | -4425.2 | 74.6 | -2.8222 | -2.44E-09 |
| C09G | 2.314 | 119.343 | -4711 | 3.2 | 2.0439 | 1.30E-09 |
| C08F | 1.597 | 119.560 | -2700.1 | 26.72 | -6.7929 | -4.33E-09 |
| M01G | -1.016 | 118.443 | -2150 | 59.5 | -55.6767 | -3.63E-08 |
| M02F | -1.189 | 118.925 | -2050.8 | 62.9 | -8.0046 | -5.23E-09 |
| M03F | -0.674 | 119.110 | -2150.1 | 62.2 | -7.4272 | -4.86E-09 |
| TOLI2 | 1.121 | 120.794 | 86 | – | – | – |
| BKB | -1.107 | 116.904 | 110 | | | |





## 2.2 Receiver functions method

The teleseismic P-wave receiver function is an effective method to obtain the crustal structure beneath each seismic station (Langston, 1977). It is a time series obtained by deconvolution of the vertical component (Z) from the radial component (R) of the teleseismic waveform, which isolates converted and reflected wave responses generated by velocity discontinuities beneath the station. For the calculation of receiver functions, teleseismic events with magnitudes greater than 5.5 and epicentral distances from 30° to 90° are employed. Using the catalogue of the International Seismological Centre (ISC), we obtain 316 seismic events from magnitudes 5.5 to 8.0 Mw (Fig. 2a).

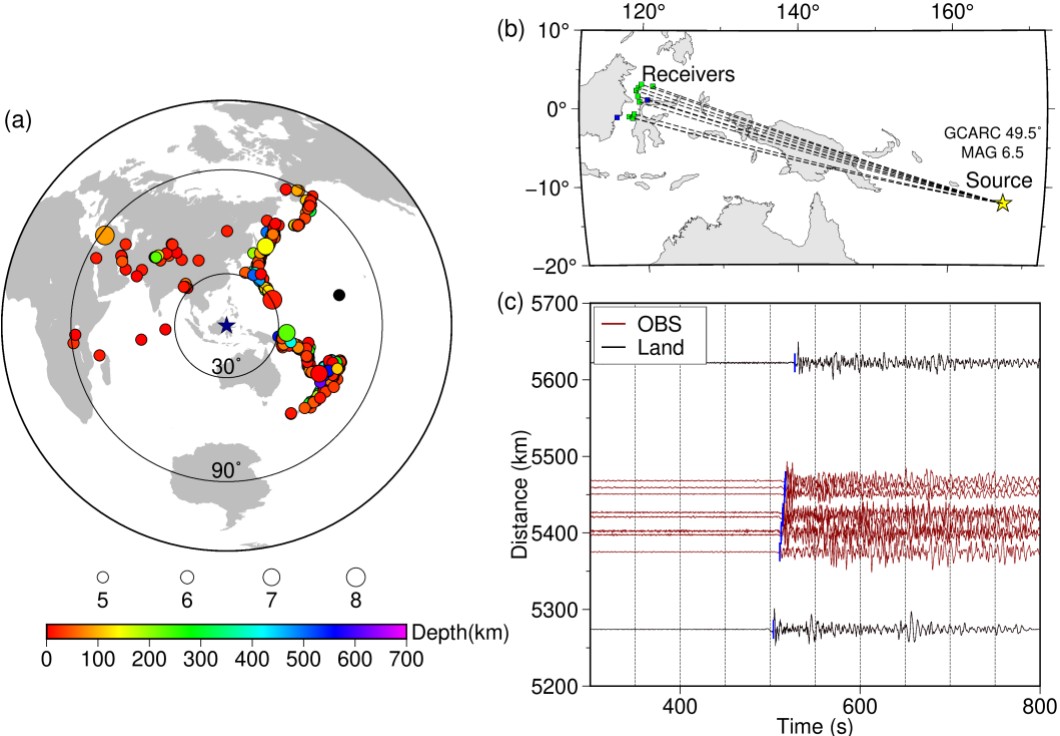

**Figure 2: (a) Distribution of teleseismic earthquake events (The circle diameter and colour represent the magnitude and depth of the sources, respectively) (b) Teleseismic propagation path in map view; (c) Teleseismic waveforms (black is from a land station ,red is from an OBS, blue represent theoretical travel time).**

Receiver functions were computed from pre-processed three-component waveforms in two main steps: coordinate rotation and time-domain deconvolution. Each event was windowed from 10 s before to 120 s after the predicted P-wave arrival, estimated using the AK135 velocity model (Kennett et al., 1995). We removed the mean and trend, applied a bandpass filter (0.02–2.0 Hz), and rotated the data from ENZ to RTZ coordinates based on event and station geometry.





Deconvolution was stabilised using a Gaussian filter with a width of 2.5 (corner frequency ~1.0 Hz). Receiver functions were also computed with Gaussian widths of 1.0 to 3.0 to assess frequency-dependent crustal response, but we focus primarily on
125   the Gaussian 2.5 results (Ligorria and Ammon, 1999). In total, 314 high-quality receiver functions were obtained (Fig. 3a, 3b), with manual selection guided by the quality control criteria of Shen et al. (2013).

## 3 Result

### 3.1 Receiver Function H–κ Stacking

The H–κ stacking technique is a widely used method to estimate crustal thickness (H) and average crustal Vp/Vs ratio (κ)
130   beneath seismic stations, under the assumption of relatively simple crustal layering (Zhu and Kanamori, 2000). The method utilises the differential arrival times and amplitudes of the direct Ps phase and its crustal reverberations—PpPs and PsPs + PpSs—generated at the Moho discontinuity. By stacking the amplitudes of these phases across a grid of H and κ values, we identify the optimal crustal parameters at the point of maximum stacking coherence. The H–κ stacking technique is a widIn this study, we adopt stacking weights of 0.5, 0.3, and 0.2 for Ps, PpPs, and PsPs + PpSs phases, respectively, following standard
135   practice to enhance signal sensitivity while suppressing noise. Example H–κ stacks for stations TOLI2 and M03F are shown in Figures 3c and 3d, while the complete results for all nine OBSs and one land station are displayed in Figure 3. The detailed H–κ stacking results of all OBSs and land stations are also presented in the Fig. S12.





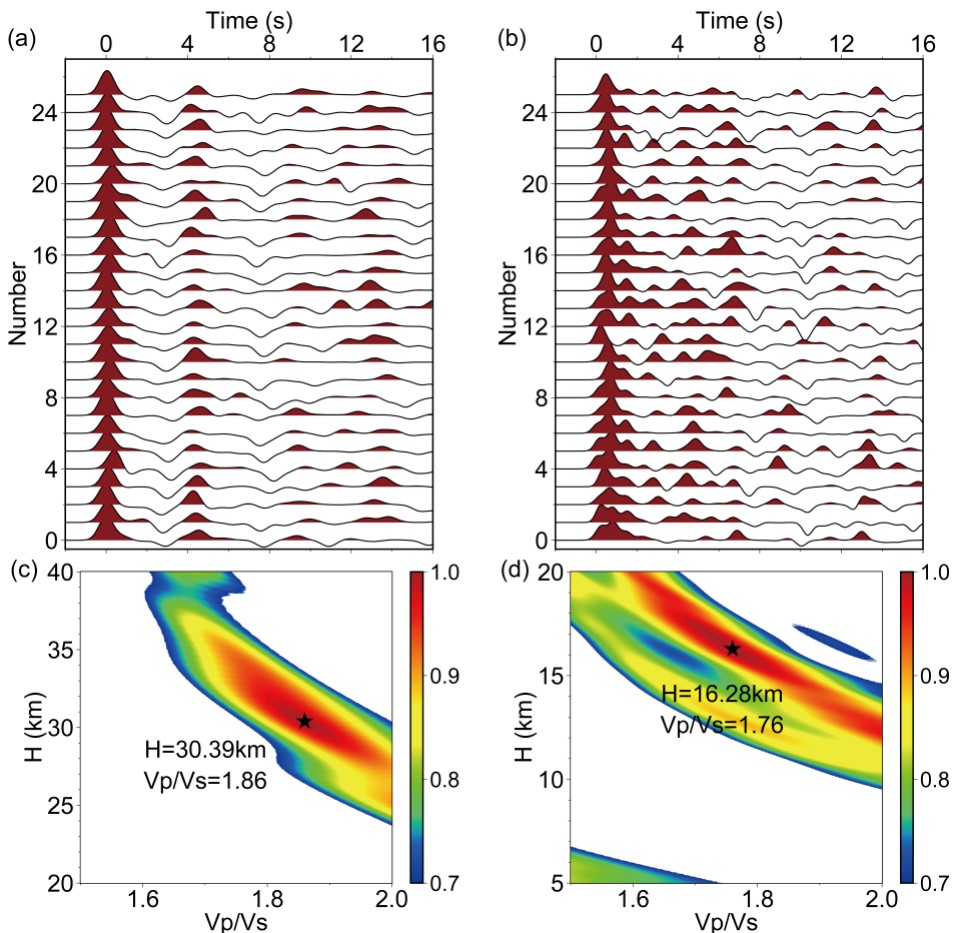

**Figure 3: (a) Receiver functions at the land station TOLI2; (b) Receiver functions at the OBS site M03F; (c) H-κ stacking result of TOLI2; (d) H-κ stacking result of M03F.**

Our analysis reveals substantial lateral variation in crustal thickness across the study region. Beneath the Celebes Sea, Moho depths are generally shallow (~8 km), consistent with an oceanic-type crust. In contrast, crustal thickness beneath the Makassar Strait and northern Sulawesi reaches ~25 km, characteristic of extended continental or transitional crust. These contrasts delineate significant tectonic segmentation across the Palu–Koro Fault system.

We also observe a general anti-correlation between crustal thickness and surface topography, broadly consistent with isostatic compensation. Moreover, the offshore Vp/Vs ratios tend to be higher, which may reflect the presence of water-saturated unconsolidated sediments or mafic compositions in the lower crust. These variations provide further evidence for complex crustal architecture influenced by active tectonism and sedimentation.



**Table 2: OBS and its adjacent land station resultant crustal thickness H, Vp/Vs and Moho depth measurements with their standard deviations.**

| Station | Lat (°) | Lon (°) | Depth (m) | H (km) | Vp/Vs | Moho(km) |
|---------|---------|---------|-----------|--------|-------|----------|
| C28F | 0.871 | 119.766 | -2400.7 | 22.70±2.95 | 2.13±0.18 | 25.10 |
| C12F | 2.689 | 119.631 | -4800.4 | 11.60±1.08 | 2.17±0.19 | 16.40 |
| C21F | 2.862 | 121.461 | -4500.4 | 7.83±0.53 | 1.82±0.11 | 12.33 |
| C18F | 3.112 | 120.001 | -4425.2 | 10.75±0.84 | 2.20±0.16 | 15.18 |
| C09G | 2.314 | 119.343 | -4711 | 12.16±0.74 | 2.24±0.07 | 16.87 |
| C08F | 1.597 | 119.560 | -2700.1 | 14.62±0.91 | 2.23±0.13 | 17.33 |
| M01G | -1.016 | 118.443 | -2150 | 23.80±0.80 | 2.10±0.07 | 25.95 |
| M02F | -1.189 | 118.925 | -2050.8 | 23.97±0.60 | 2.29±0.06 | 26.02 |
| M03F | -0.674 | 119.110 | -2150.1 | 22.65±0.44 | 2.14±0.03 | 24.80 |
| TOLI2 | 1.121 | 120.794 | 86 | 30.39±1.37 | 1.86±0.055 | 30.30 |

## 3.2 Receiver Function Waveform Inversion

To extract more detailed S-wave velocity profiles and assess vertical crustal heterogeneity, we perform waveform inversion of receiver functions using a neighbourhood algorithm (NA) approach (Sambridge, 1999a). This nonlinear global optimisation method allows exploration of high-dimensional model spaces without assuming a priori smoothness or linearity. The inversion is carried out in two stages. First, we randomly sample 2000 velocity models over the entire parameter space and compute their misfits to observed receiver functions. Misfit is quantified as the least-squares difference between observed and synthetic receiver functions, computed using reflectivity-based forward modelling. The ten best-fitting models are then used as seed points to generate 40 new models each via stochastic perturbation within Voronoi cells, iterating this process for 200 generations. This strategy ensures both global coverage and local refinement in the model space (Sambridge, 1999b).

Example results for OBS stations C28F and M03F are presented in Figure 4. The synthetic receiver functions derived from the optimal velocity models exhibit good agreement with the observed data, supporting the robustness of the inversion. The detailed fitting results of all OBS receiver function waveforms and the 1D S-wave velocity models are also presented in the supporting information (Fig. S6-S11). Notably, the S-wave velocity structure reveals a clear Moho step between southern M03F and northern C28F, suggesting a tectonic boundary or crustal offset across this region (Fig. 4a, b). The complete set of inversion results is summarised in Figure 6c. The inferred Moho depths vary markedly across the network, with significant offsets observed across the Palu–Koro Fault zone. This crustal discontinuity provides strong geophysical evidence for block juxtaposition and active fault-related deformation beneath the Celebes Sea region.





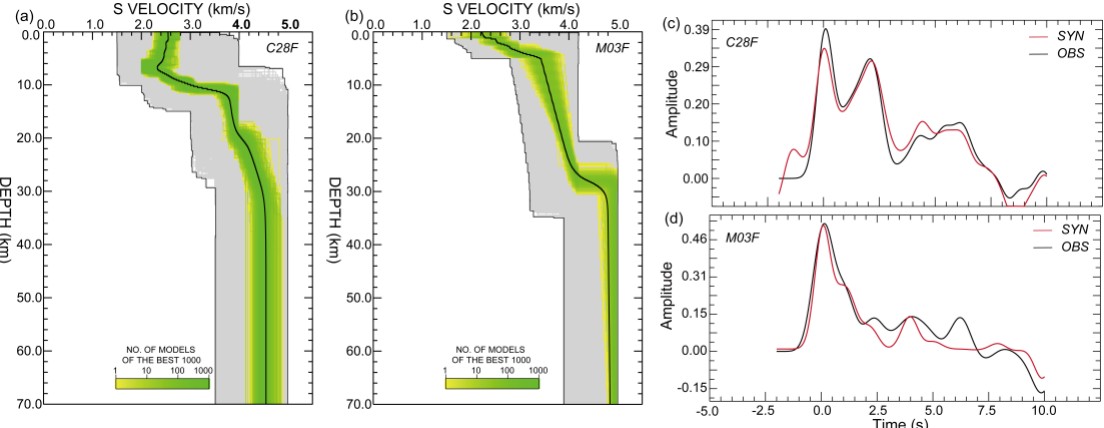

Figure 4: Receiver function waveform inversion result at OBSs site C28F and M03F. (a) and (b) The S-wave velocity structure obtained by the NA inversion (The thin black line encloses the search area and the black curve is the average velocity structure of the 1000 best models). (c) and (d) Receiver function waveform comparison (The black line is the observed receiver function and the red line is the receiver function predicted by the optimal model).

## 4 Discussion

### 4.1 Crustal Structure Characteristics in the Sulawesi Region

Our H–κ stacking results reveal significant lateral variation in Moho depth across the Sulawesi region, with particularly pronounced changes in the east–west direction across the Makassar Strait (Fig. 5a). The Moho beneath oceanic domains, such as the Celebes Sea, is relatively shallow, with depths averaging ~25 km, while the crust beneath northern Sulawesi reaches thicknesses exceeding 30 km. Notably, the island crust is at least 5 km thicker than the central Makassar Strait crust. This result is consistent with previous studies (e.g., Greenfield et al., 2021), although our broader dataset and additional constraints lend greater robustness to the observed variations.

We interpret these results as evidence that continental crust dominates the region between Sulawesi and Borneo, particularly in the western part of the Makassar Strait, whereas crustal thinning and oceanic affinity appear more prominent toward the Celebes Sea. The sharp crustal contrast suggests the Makassar Strait marks a crustal boundary zone, with implications for the tectonic assembly of Sulawesi. In particular, our findings support the hypothesis that Sulawesi accreted to the eastern margin of Sundaland during the Late Miocene (Hall and Wilson, 2000), potentially along a pre-existing crustal suture.

A gradual thickening of the crust is also observed from the North Sulawesi Trench landward toward the northern arm of Sulawesi. This gradient may reflect differences in the nature and evolution of the overriding plate. Previous studies have proposed that the overriding plate in this region has experienced significant extensional deformation since the Miocene, as evidenced by the widespread distribution of Cenozoic normal faults in northern Sulawesi (Hall, 2012; Advokaat et al., 2017;



Zhang et al., 2018). Despite evidence of active subduction along the North Sulawesi Trench from the Late Pliocene to Holocene, there is limited geologic evidence for compressional tectonics onshore (Advokaat et al., 2017; Hall, 2019).


Recent numerical modelling (Song et al., 2022) suggests differential trench rollback along the North Sulawesi Trench may explain the observed rotation of northern Sulawesi, and our Moho depth variations across this zone are consistent with such differential tectonic behaviour. We suggest that the ~30 km Moho depth beneath northern Sulawesi reflects its origin as an ancient island arc terrane, while the significantly thinner crust (~15 km) beneath the Celebes Sea represents subducting oceanic

lithosphere. The ~19 km Moho depth contour at ~120°E and 0°–2°N marks a north–south trending transition zone between the Makassar Strait and northern Sulawesi. This feature aligns spatially with the Palu–Koro Fault zone, providing supporting geophysical evidence for its role as a major tectonic boundary separating the Sunda block to the west from micro blocks associated with the Indo-Australian Plate to the east (Socquet et al., 2019; Spakman and Hall, 2010).

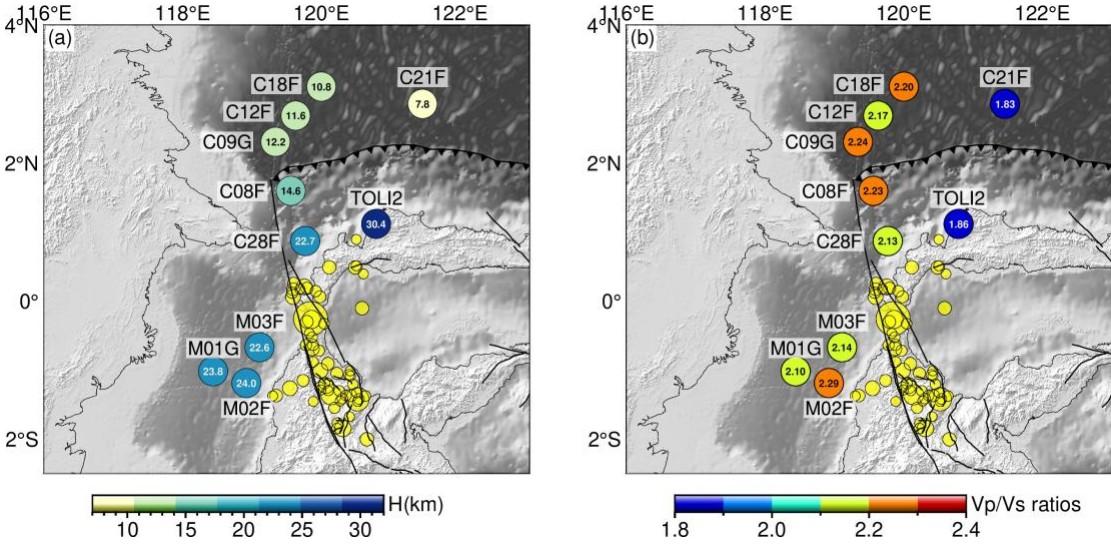

**Figure 5: (a) Crustal thickness in the study area; (b) Vp/Vs ratios in the study area. (The red solid line represents the Palu-Koro fault (Patria and Putra, 2020), and the yellow solid circles represent the epicentre distribution of the earthquake after relocation)**

### 4.2 Velocity Structure and Seismicity Along the Palu–Koro Fault

The Palu–Koro fault is a major left-lateral strike–slip fault that originated during the Paleogene (ca. 65–23 Ma), transecting Sulawesi and effectively dividing the island into northern and southern tectonic domains. Presently, the northern arm of

Sulawesi undergoes clockwise rotation at a rate of approximately 4°/Ma, largely driven by the combined influence of subduction along the North Sulawesi Trench and strike–slip displacement along the Palu–Koro fault (Hall, 2019; Lü et al., 2019). Despite the fault's high slip rate, the historical record indicates a surprisingly low frequency of large earthquakes (Bellier et al., 2001).



On 28 September 2018, a Mw 7.5 earthquake ruptured the fault near Palu city, producing a devastating tsunami with run-ups exceeding 11 meters (Song et al., 2019). Back-projection and aftershock studies indicate rupture of at least two fault segments, including a submarine portion with near-ideal strike–slip behaviour. The fault appears to accommodate both trench retreat and oblique crustal deformation, which may help explain the complex rupture characteristics and tsunami generation (Song et al., 2019).

Thus, several key questions remain unresolved, such as, where the submarine segment of the Palu–Koro fault lie relative to the seafloor, why this fault accumulated such significant strain with so few large seismic events and what factors contributed to the sudden onset of the 2018 rupture.

To better constrain the offshore fault geometry and rupture dynamics, we employed a two-pronged approach: relative
aftershock relocation and back-projection of the mainshock rupture. We relocated all Mw ≥ 4.0 aftershocks using waveform data from 817 broadband stations of the China Array network. After preprocessing with a bandpass filter (0.5–2.0 Hz) and correcting for station terms, we performed a grid search over a 40×40 node space (5 km spacing) in the source region, using 10-second time windows offset by 1 second. The location corresponding to the maximum stacking amplitude of P-wave arrivals was selected as the most probable hypocenter.


Back-projection was also used to track the mainshock rupture front. By aligning the high-frequency P-wave energy and identifying the grid nodes that produce the strongest coherent stacking (Wang et al., 2018), we traced the rupture propagation from the hypocenter northward. Our results suggest that rupture rapidly advanced toward the NNW, reaching the northern fault segment shortly after the initial rupture, and propagating southward with a ~20 s delay (Fig. 6b). This asymmetric rupture
behavior is consistent with field-mapped fault traces on land and their inferred offshore extensions (Hall and Wilson, 2000; Hamilton, 1979; Rangin et al., 1999).

Additionally, we integrated our receiver function inversion results to characterize the S-wave velocity structure across the fault zone. The interpolated S-velocity profile (Fig. 6c), taken along the cross-section marked by the dashed white line in Fig. 6a,
reveals a prominent low-velocity anomaly extending beneath OBS C28F in the Celebes Sea. This feature, interpreted as a zone of fractured or partially serpentinized material, aligns spatially with the fault trace inferred from seismicity.

Low-velocity anomalies or abrupt lateral velocity gradients are commonly associated with fault zones in seismic imaging studies (Thurber et al., 1997). The presence of such a feature at mid-to-lower crustal depths supports the interpretation that the
Palu–Koro fault extends through the crust and offsets the Moho discontinuity. The geometry of the anomaly suggests that the fault dips gently with increasing depth, characteristic of a supracrustal submarine fault system. Together with the relocated aftershocks, these findings provide strong evidence that the Palu–Koro fault is a deep, throughgoing structure that accommodates both strike-slip motion and vertical crustal deformation offshore.





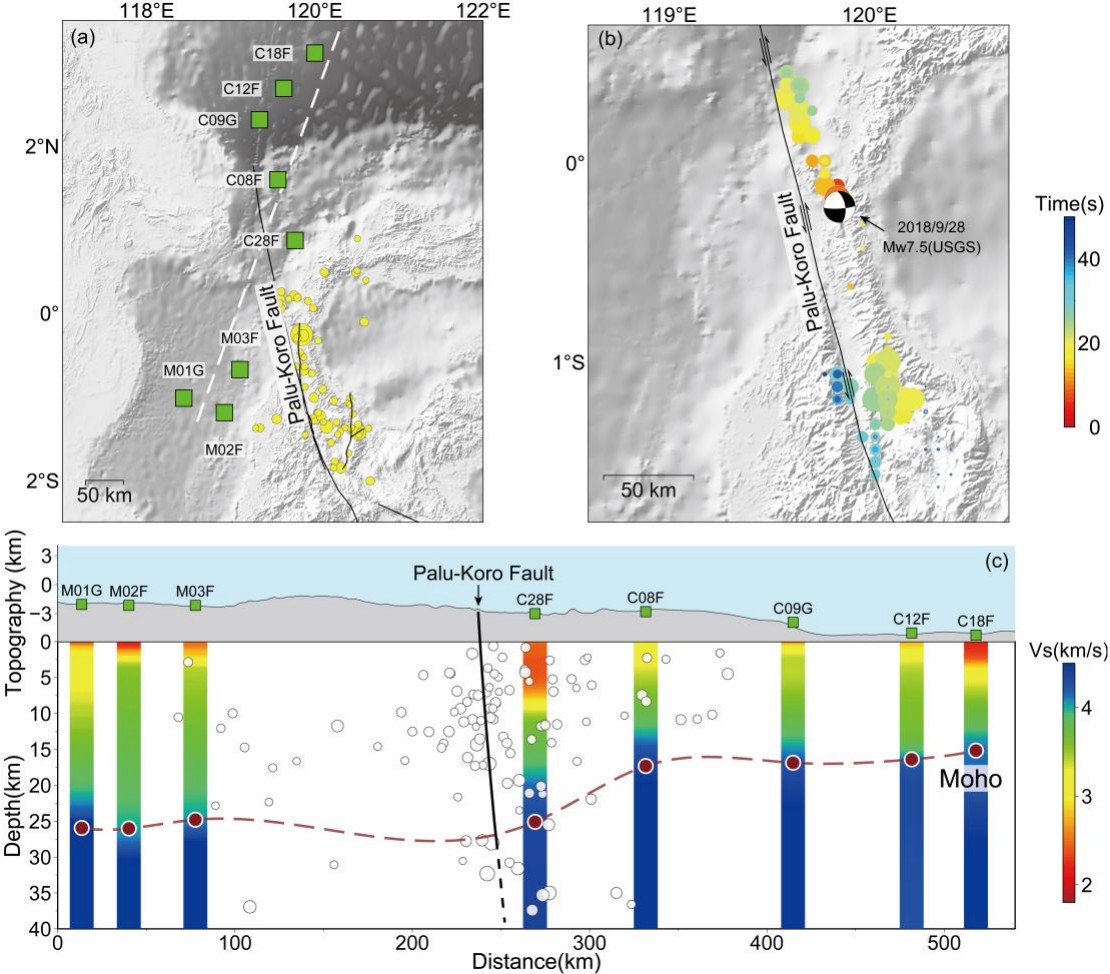

**Figure 6: S-wave velocity structure and seismicity near the Palu–Koro fault. (a) aftershocks distribution after relocation (The black solid line represents the location of the Palu–Koro fault, and the yellow solid circles denote the aftershocks). (b) Back projection by the Mw 7.5 Palu earthquake (The size of the circle indicates the magnitude of the earthquake, while the colour indicates the origin time of aftershocks after the mainshock). (c) The S-wave velocity structure near the Palu–Koro fault (The velocity structure is obtained by receiver function inversion results. The white circle represents M > 3 earthquakes and the black line is the Palu–Koro fault; The red line indicates the Moho depth obtained by the H-κ stacking results).**

## 5 Conclusion

In this study, we present new insight into the crustal structure and tectonic framework of the Sulawesi region based on nine OBSs and two land stations. Using receiver function analysis and waveform inversion, we obtain the crustal thickness, Vp/Vs ratios, and S-wave velocity below the OBSs and land stations in the Celebes Sea and Makassar Strait area. The results show that:



A pronounced lateral variation in crustal thickness: the crust beneath the Celebes Basin is relatively thin (~8 km) with gradually thickened sediment layer toward east as much as ~15km, consistent with previous study of the oceanic crust thickness from the wide-angle refraction study in the 90s, whereas crustal thickness increases significantly toward the Makassar Strait (~25 km) and reaches over 30 km beneath North Sulawesi. These findings suggest that the Makassar Strait may be underlain by transitional or modified continental crust, rather than purely oceanic. The elevated Vp/Vs ratios offshore may reflect the presence of water-saturated sediments or fractured crustal material.

It appears that the two sides of the fault represent separate crustal blocks. The H–κ stacking and receiver function inversion results provide robust constraints on the geometry of the Moho across the study area, revealing uplifted crustal blocks and sharp crust–mantle transitions. Based on an analysis of the S-wave velocity structure and seismicity, the NNW extension of the Palu–Koro fault into a offshore region along this fault zone indicates a deep-reaching, low-velocity zone that likely corresponds to a crustal-scale strike–slip system., and it is supported by the distribution of relocated aftershocks and back-projection analysis of the 2018 Mw 7.5 Palu earthquake, which reveal rapid rupture propagation and segmentation along the fault.

This study suggests that the Palu–Koro fault is a major lithospheric boundary that facilitates both lateral shear and vertical deformation in response to regional tectonic forces. Our findings support the hypothesis that the fault accommodates oblique convergence and trench retreat processes, and that the crustal architecture of Sulawesi is shaped by the interaction of microplate fragments within the broader Indo–Australian–Eurasian plate system.

More seismological investigation to refine models of crustal evolution should help to give an in-depth understanding of the interaction and seismo-tectonic hazard in this complex region and helps to explain the phenomena of earthquake occurrence along extensive strike slip faults.

## 6 Acknowledgments

The work contained in this publication was supported by NSFC Major Research Plan on West-Pacific Earth System Multispheric Interactions (project number 91858212) and NSFC general program (project number 42276072). This study was funded by a University of Cambridge GCRF (project number G102642), and CL was funded through the Isaac Newton Trust and International Partnership of the Chinese Academy of Sciences (project number 132A11KYSB20180020).

**Code/Data availability**

Seismic waveform data from ocean-bottom and land stations are available upon reasonable request due to institutional agreements. Processed data and modelling results are available from the corresponding author. The hk1.3 software package was used for calculating receiver functions and performing H-κ stacking (https://www.eas.slu.edu/People/LZhu/home.html).





The Neighbourhood Algorithm na_sampler package was used for receiver function inversion

(https://iearth.edu.au/codes/NA/NA_sampler.php).

**Author contributions**

T.Y did the data analysis and manuscript writing. C.L. led the research and the fieldwork, did the data curation and contributed to the data analysis and assisted with manuscript writing. T.Y and N.R. secured funding and supported the fieldwork. T.X. contributed to inversion and interpretation. A.A., M.T.R. and D.P.S. supported fieldwork and data recovery.

All authors reviewed and approved the final manuscript.

**Competing interests and licences**

The authors declare no competing interests. All reused content is properly cited, and licensing terms are stated where applicable.

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
