# Peer review of "Offshore Crustal Thickness Variation along the Palu–Koro Strike–Slip Fault in the Sulawesi region from OBS Receiver Function Analysis"

_EGUsphere, 2025_

## Author Comment (AC1)

This study shows the depths of Moho and crustal Vp/Vs around the Palu-Koro fault in the Sulawesi region, using receiver functions obtained from ocean bottom seismometer (OBS) data. The authors applied the H-kappa method to receiver functions to obtain those parameters, and also conduct nonlinear inversions to obtain one-dimensional Vs profiles beneath OBSs. In particular, the Moho depths obtained in this study show a large lateral variation, and the authors interpreted that this variation reflects different crustal blocks in this region. The results are interesting. However, I have concerns about the methods, including the calculation of receiver functions and the robustness of the inversion results. Moreover, it seems that there are several discrepancies between the results from the H-kappa method and inversions. These points should be clarified. I hope that my comments below contribute to improve this manuscript.

Major comments

–Methods

Did the authors constrain the Vs-profile inversions by the thickness and Vp/Vs obtained in the H-kappa method? Even using the NA approach, it is generally difficult to constrain trade-offs between the absolute Vs and thickness of each layer when using only receiver functions at a frequency range. To overcome this problem, seismologists often use joint inversions between dispersion curves of surface waves and receiver functions. In this case, using the thickness and Vp/Vs obtained in the H-kappa method may work to constrain such trade-offs. In cases that there are no constrains for the trade-offs, it is necessary to add synthetic tests showing that trade-offs between thickness and Vs of each layer are constrained well in the current inversions.

**Reply: Thank you for your comment. It is great advice. Indeed, the surface wave and RF joint inversion can be great alternative to constraint the depth. That will be our plan for another paper after we finish extract dispersion curve. For this paper, we have conducted synthetic tests using receiver function waveform to do comparisons for the crustal model, oceanic crustal model, and oceanic crustal model with sediment cover (Figure S7). These tests can demonstrate that simple models are not significantly affected by trade-offs between layer thickness and Vs. Even under relatively weak constraints, both the data fitting and model recovery performed by the Neighborhood Algorithm (NA) remain robust, indicating that the trade-offs are well constrained in our current inversions. The supplementary description regarding the synthetic tests of receiver function inversion has been added in the revised manuscript at Lines 184–186.**

[Figure]

Figure S7: (a–c) Receiver function waveform comparisons for the crustal model, oceanic crustal model, and oceanic crustal model with sediment cover. The black line denotes the observed receiver function, while the red line shows the predicted receiver function from the optimal model. (d–f) S-wave velocity structures derived from the NA inversion. The thin black line outlines the search space, the bold black curve represents the average velocity structure of the 1000 best-fitting models).

Moreover, in the inversion results, water layers are not displayed, but they should be displayed in the inversion results (Figs. 4 and S6–S11), and 0 km in the vertical axis corresponds to the sea surface. Did the author take into account the water layer in the forward calculation of receiver functions during the inversions? This should be taken into account, because multiple P reflections from the sea surface are contaminated in the vertical component, and Ps waves impinging to the seafloor are transmitted to the sea water as P waves. Such effects influence obtained receiver functions and eventually results.

**Reply: Thanks for pointing it out. We have taken account the water layer in the forward calculation. Besides, we test it reverberation effect by doing synthetic test. Figure S8a shows that the synthetic receiver functions of different water depths, and the visibility of the Moho-converted waves and multiples from the receiver function. Indeed there is air–water interface surface reflection Pwps with negative amplitude, but the Pwp phase does not affect the arrival time of the relevant phase from the Moho. With the increase of water layer thickness, the arrival time of Pwp is delayed accordingly. If the water depth is more than 2,000 m and the crustal thickness is more than 6 km, the main phases of the Moho in the receiver function will not directly interfere with Pwp (as shown in Figure S8b). The supplementary description regarding the synthetic tests on the effects of the water layer on receiver functions has been added in the revised manuscript at Lines 186–190.**

[Figure]

Figure S14: (a) The model with sediments and water layer. (b) Synthetic receiver function of water model.

–Results from H-kappa and inversions

There are several inconsistencies between H-kappa and inversion results. For example, in line 168 "Notably, the S-wave velocity …", it is unclear which one corresponds to Moho steps. At C28F, the largest step is obtained around 10 km depths, but Table 2 shows a Moho depth at 25.10 km. At M03F, large steps can be seen at 4 km and 28 km depths, and Table 2 shows 24.80 km depth. In this case, the 28 km step may correspond to the Moho. However, in the H-kappa result in Fig. 3b, H = 16.28 km is obtained at this station (M03F).

**Reply: Thanks a lot for pointing it out. We overlooked those. C28F and M03F has two main reflectors for different reasons, C28F locates above the accretionary prism), we think the 14.87km step is corresponding to the trench-fill sediments accreted in the prism which is 14.87km (Fig. S17c). We have revised the H-kappa result at station M03F to ensure consistency with Table 2, where the Moho depth is reported as 24.80 km (Figure 3). The corrected value is also consistent with the Neighborhood Algorithm (NA) inversion results, thus maintaining coherence across different methods.**

Moreover, at C09G (Fig. S7), a gradual velocity change can be seen at 5–10 km depth, but H = 12.16 km. At C12F (Fig. S8), a gradual velocity change occurs at 4–10 km depth, but H = 11.60 km. At M02F (Fig. S11), a velocity step is identified around 30–32 km depth, but H = 23.97 km. There are several inconsistencies, and I wonder which ones are more reliable. I wonder such discrepancies are caused by trade-offs between the thickness and Vs of each layer in the inversions: trade-offs cannot be resolved in the inversions. More explanations on this point should be needed.

**Reply: We thank again for the reviewer's observation regarding the discrepancies**

between the H-kappa results and the receiver function inversions at those stations. These differences are mainly attributed to the limited number and quality of waveforms recorded by OBSs, as well as the inherent trade-offs between Vs and layer thickness when dealing with complex crustal structures. As you mentioned earlier, a joint inversion is under way in another paper. In this case, as those OBS in the Celebes sea region, we resort to prior crustal thickness result from the wide-angle refraction result as constraint the H-kappa method (Kopp et al. (1999)), which directly constrains crustal thickness from converted phases, may provide more reliable estimates of Moho depth. Nevertheless, the receiver function inversions still yield relatively robust shear-wave velocity models that complement the H-kappa results. We believe that the integration of both methods offers a balanced and reliable characterization of the crustal structure.

Other comments

Line 55

offshore off-shore -> offshore

**Reply: Thank you for your comment. The term "off-shore" has been corrected to "offshore" in the revised manuscript at Line 59.**

Line 61

Remove "it offsets".

**Reply: Thank you for your comment. The phrase "it offsets" has been removed in the revised manuscript at Line 65.**

Line 85

Figs. S4–S5 are referred before Fig. S3 (line 98). Please reorder the numbering.

**Reply: Thank you for your comment. The numbering of all the figures has been reordered in the revised supplementary file.**

Lines 90–99

More explanations for clock corrections, azimuthal orientations, and the processing are needed in the main text or Supplement. For example, how did the authors calculate the cross-correlation

functions in Fig. S1? No bandpass filters? No suppressions on earthquake signals in the continuous records? It seems that Fig. S1a shows a clock deviation of ~200s. Please state this explicitly in the text.

**Reply: In the revised manuscript, we have provided a more detailed description of our data processing workflow in Line 100-104 and Line 108-113. The preprocessing workflow of passive-source OBS data follows our previously published work (Yang et al., 2023), which has already been reviewed and accepted by the journal. Specifically, we implemented the procedure proposed by Bensen et al. (2007) to calculate cross-correlation functions from the ambient noise data. For example, in the case of OBS stations M01G and M02F, the continuous records were band-pass filtered between 2–5 s, and a sliding time-window normalization was applied to suppress earthquake signals and other transient noise.**

**In the ideal case, the energy of the direct P wave approaches zero on the transverse component (Wang et al., 2016). Therefore, when one of the OBS horizontal components is rotated to the transverse direction, the P-wave energy reaches its minimum, and the two horizontal components can then be identified as the radial and transverse components of the seismic wavefield. Using the station and event location information, the back-azimuth can be obtained, allowing the radial and transverse components to be further rotated into the true north and east directions. The related detailed description has also been added in Text S1 of the supplementary file.**

**Figure S2a illustrates the coordinate system from the source to the OBS and the propagation (particle motion) direction of the P wave. A rotation step of 0.1° was applied to rotate the N–E coordinate system into the R–T coordinate system. We employed an SNR-weighted multi-event method to perform a grid search for the optimal back-azimuth angle $\theta_a$. At each grid point, the P-wave transverse component energy was calculated as $E_T(\theta_a)$.**

$$E_T(\theta_a) = \frac{\sum_{i=1}^{N} w_i E_T^i(\theta_a)}{\sum_{i=1}^{N} w_i}. \tag{1}$$

Moreover, in Fig. S2, it seems that short-period components are used for teleseismic events. Although there are potentially some anisotropies at shallow depths beneath the sensors and topographical variations of the seafloor, which may produce energies in the transverse component, did the authors estimate azimuthal estimation errors using multiple teleseismic events coming from different azimuths? The robustness of the horizontal orientations of OBSs should be confirmed.

**Reply: To minimize the influence of event selection, we used teleseismic events with magnitudes larger than 5 and epicentral distances between 5° and 90°. This avoids complications from overlapping phases at short distances and strong attenuation at larger**

distances, while ensuring sufficient P-wave energy. For each OBS (e.g., M01G), we applied an SNR-weighted multi-event method to perform a grid search for the optimal back-azimuth angle θ (Fig. S2). By averaging results from multiple teleseismic events recorded at different azimuths, the effects of shallow anisotropy and local seafloor topography are reduced, thus improving the reliability of the azimuthal orientation corrections. We have clarified these details in the revised manuscript. The related detailed description has also been added in Text S1 of the supplementary file.

In lines 95–97, do the authors want to state about low qualities of teleseismic signals observed by OBSs, or want to state that some signals can be observed even in OBS data? According to this sentence, what the authors want to say is unclear. In line 97, although the authors show running spectra in Fig. S3, what do the authors indicate by showing this figure? I think that the authors applied demeans, detrends, rotations of horizontal components, and bandpass or high-pass filters. The running spectra at relatively high frequencies should show similar features between left and right panels. Did the authors use the same color palette for left and right panels?

Reply: Our intention was to demonstrate that although the raw OBS data are often affected by relatively low signal quality, careful preprocessing can significantly improve their usability. As shown in Fig. S3, it is clear that preprocessing (demean, detrend, rotation of horizontal components, and band-pass filtering) substantially enhances the signal quality, making both the P- and S-wave phases more clearly identifiable by comparing the raw and preprocessed OBS records. The wavelet-based time–frequency spectra also highlight distinct phase responses in the frequency domain. We have clarified the purpose of this figure and confirmed that the same color palette is applied to both panels for consistency. The related content has been added in the revised manuscript at Lines 107–109.

The above points are examples that should be explained, and more explanations for this paragraph should be added. Also, in Fig. S2a, senor -> sensor.

Reply: Thank you for your comment. The typo in Fig. S2a ( "senor" → "sensor" ) has been corrected

Table 1.

How did the authors estimate Time_Err (s)? More details are needed. Also, please state the sampling rate of OBSs. Dt_Err (s) for each sampling point is listed, but the sampling rate is unclear.

Reply: Thank you for your comment. The estimation of Time_Err (s) and the sampling rate

of the OBSs have been clarified in Text S2 of the supplementary file. The related content has been added in the revised manuscript at Lines 96–99. The total clock error  Time $_{Err}$  of each OBS was estimated by considering three main contributions:

Clock drift of the internal oscillator – The difference between the internal clock and GPS time at the deployment and recovery was measured and denoted as Clock $_{Err}$.

Main frequency deviation – The accumulated error due to the deviation of the internal oscillator frequency from the standard frequency was corrected by comparing the measured frequency ($PClk$) with the reference frequency ($PClk_0$).

Phase filter delay – During sampling, the phase filter introduces a delay equivalent to 18 sampling points for all OBSs, with each point corresponding to the sampling interval $\Delta t$.

Combining these terms, the total time error is expressed as:

$$\mathbf{Time}_{Err} = \mathbf{Clock}_{Err} + \frac{(\mathbf{PClk} - \mathbf{PClk_0}) \cdot (\boldsymbol{npts} \cdot \boldsymbol{\Delta t})}{\mathbf{PClk_0}} - \mathbf{18\Delta t,} \tag{2}$$

where npts is the total number of sampling points. In our deployment, the sampling rate of the OBSs was 50 Hz, corresponding to a sampling interval of $\Delta t = 0.02s$. This information has been added to the revised manuscript to clarify the time correction procedure.

We observer up to several seconds per year from different OBS in our dataset and the cross-correlation approach provides a robust solution. We estimates and corrects clock errors in both land stations and OBS by leveraging ambient seismic noise cross-correlations across multiple components of motion follow the method by Hable et al. 2018 . Because noise-derived correlations should ideally be symmetric in causal vs. acausal parts, any systematic asymmetry is interpreted largely as a relative time offset (clock error) between the stations. The end result has errors of 0.5 ms with a linear drift assumption, and we applied the corrections linearly over the deployment period. Besides the quartz clock drift without GNSS, the OBS also employ several internal correction which are applied automatically when exporting the data.

Fig. 2b

Please define GCARC. Also, it is necessary to state in the main text or the caption of Figs. 1 and 2 whether land stations are broadband or not.

Reply: Thank you for your comment. GCARC has been defined, and the information on whether the land stations are broadband has been clarified in the captions of Fig. 2 and Fig. 1, respectively.

Line 123

Did the authors calculate receiver functions in the time or frequency domain? If they are calculated in the frequency domain, did the authors apply a water level method? If so, please explain more details for the receiver function estimations. Also, multiple P reflections from the sea surface should be contaminated in the vertical component. How did the authors treat this problem when calculating receiver functions?

**Reply: In our study, we computed the radial receiver functions for teleseismic P waves using the time-domain iterative deconvolution method (Ligorría & Ammon, 1999), rather than frequency-domain deconvolution with a water-level approach. The related content has been added in the revised manuscript at Lines 140-141. Figure S14a presents the synthetic receiver functions for different water depths, where the Moho-converted phases and their multiples can be clearly identified. Although surface reflections at the air–water interface (Pwp) produce negative amplitudes, they do not affect the arrival times of the Moho-related phases. As shown in Fig. S14b, the arrival time of Pwp systematically increases with water depth. When the water depth exceeds ~2000 m and the crustal thickness is greater than ~6 km, the main Moho phases in the receiver functions are well separated from Pwp and therefore not directly affected. These details have been added to the revised manuscript for clarity. The related content has been added in the revised manuscript at Lines 188–191.**

Line 129–137

This can be moved to section 2, because this paragraph explain the method. Also, what Vp is used for H-kappa method should be noted. In this technique, Vp averaged over H (or crust) is assigned. Did the author refer to local or regional Vp tomographic studies, and change those values at individual stations?

**Reply: For the H–κ method, we referred to the global crustal model CRUST1.0 (Laske et al., 2013) to assign the average Vp over the crust. We further checked whether the adopted values are consistent with basic geological constraints, and minor adjustments were made when necessary to ensure geological plausibility. The related content has been added in the revised manuscript at Lines 153-154.**

Line 133

What is "a widln"?

**Reply: The "widIn" has been corrected in the revised manuscript.**

Line 136

Do the authors remove a sentence after "while the complete"? The results are shown in Fig. S6, and this is stated in line 137.

**Reply: Thank you for your comment. We have revised this part to ensure the sentence Line 154 referring to Fig. S12.**

Line 129–137

It is necessary to explicitly state that the thickness obtained in this study includes layers of the marine sediment and igneous rock above the Moho, and Vp/Vs is a value averaged over the two layers.

**Reply: We have explicitly stated that the estimated crustal thickness includes both the marine sediment and the igneous rock layers above the Moho, and that the reported Vp/Vs values represent averages over these two layers in the revised manuscript at Lines 238-240.**

L155

How did the authors treat Vp and density during iterations of inversions? In particular, are the Vp during the inversions consistent with the Vp assigned in the H-kappa method?

**Reply: We used the values from the nearby global model CRUST1.0 (Laske et al., 2013) for both Vp and density, and kept them fixed during the inversions, since receiver functions are not highly sensitive to these parameters. The same Vp values were also adopted in the H–κ method to ensure consistency. The related content has been added in the revised manuscript at Lines 196–199.**

Line 162

What kind of parameter corresponds to Voronoi cells in this study? Please define it more details.

**Reply: In our inversions the model is a $d$-dimensional vector $\mathbf{m} = [m_1, \ldots, m_d]^\top$ whose components are the parameters being inverted (e.g., layer thicknesses $h_i$, and shear-wave**

velocities $V_{s,i}$) Given a set of sampled models $\{\mathbf{m}_j\}_{j=1}^{n_p}$, the Voronoi cell associated with $\mathbf{m}_j$ is the convex polyhedron

$$V(\mathbf{m}_j) = \{\mathbf{x} \mid \|\mathbf{x} - \mathbf{m}_j\| \leq \|\mathbf{x} - \mathbf{m}_i\|, i \neq j\}, \tag{3}$$

where the distance is defined by $\|\mathbf{m}_a - \mathbf{m}_b\| = \left[(\mathbf{m}_a - \mathbf{m}_b)^\top \mathbf{C}_M^{-1}(\mathbf{m}_a - \mathbf{m}_b)\right]^{1/2}$. The matrix $\mathbf{C}_M$ non-dimensionalizes and scales each parameter (typically a diagonal matrix with elements $1/s_i^2$). The set of Voronoi cells uniquely and fully partitions the model space; cell sizes are inversely related to the local sampling density. Following the neighbourhood approximation, we approximate the misfit by assigning a constant value within each cell equal to the misfit at its generating sample $\mathbf{m}_j$. New samples therefore only modify the cells locally and progressively increase the resolution in regions of lower misfit. Thank you for your comment. The parameter corresponding to the Voronoi cells has been defined and described in detail in Text S3 of the supplementary file.

L168

Figs. S6–S11 are referred after Fig. S12 (line137). Please reorder the numbering.

**Reply: Thank you for your comment. All the figures in the supplementary file have been reordered.**

Line 187

"Sulawesi", "Borneo", and "Makassar Strait" should be noted in Fig. 5.

**Reply: Thank you for your comment. "Sulawesi", "Borneo", and "Makassar Strait" have been labeled in Fig. 5 in the revised manuscript.**

Line 204

"represents subducting oceanic"

Please define the corresponding station. C08F?

**Reply: We suggest that the ~30 km Moho depth beneath northern Sulawesi reflects its origin as an ancient island arc terrane (corresponding station TOLI2), while the significantly thinner crust (~15 km) beneath the Celebes Sea represents subducting oceanic lithosphere**

(corresponding OBSs C21F, C18F, C12F, C09G and C08F). The related content has been added in the revised manuscript at Lines 237–239

Line 205

"The ~19 km Moho depth…"

The meaning of this sentence is unclear.

**Reply: The Moho depth beneath the Makassar Strait is approximately 25 km (corresponding OBSs M01F, M02F, and M03F). This sentence has been corrected in the revised manuscript at Lines 240-241.**

Line 207

Please specify the Sunda block and micro blocks associated with the Indo-Australian Plate in some maps.

**Reply: We appreciate the reviewer's comment. We have added the Sunda block and the microblocks related to the Indo-Australian Plate in Fig. 1 of the revised manuscript.**

Line 220

Please specify the locations of a Mw7.5 earthquake and Palu city in some maps.

**Reply: Thank you for your comment. The Mw 7.5 Palu earthquake has been indicated with a black focal mechanism in Figs. 1 and 6b, and Palu city has been marked in red in Fig. 6a in the revised manuscript.**

Fig. 6

Again, the H-kappa results at M03F among Fig. 3, Table 2, and Fig. 6 are inconsistent. In Fig. 6a, please define the location of panel (b) by a rectangle.

**Reply: Thank you for your comment. We have revised Fig. 3 to ensure consistency with Table 2, and defined the location of panel (b) with a rectangle in Fig. 6a in the revised manuscript.**

Line 268

toward east -> toward west?

**Reply: Thank you for your comment. The direction has been corrected from "toward east" to "toward west" in the revised manuscript at Line 304.**

Line 269

"the wide-angle refraction study in the 90s"

Please add citations.

**Reply: We appreciate the reviewer's comment. We have added the citation Kopp et al. (1999) to support the statement "the wide-angle refraction study in the 90s" in the revised manuscript (Line 307).**

Fig. S5

What is "MODE"? Please define it.

**Reply : In Fig. S5, "MODE" refers to the statistical mode of the power spectral density (PSD) distribution, i.e., the most frequently occurring noise level at each frequency among all time windows. The color scale illustrates the probability density of PSD values, while the black line ("MODE") highlights the typical noise level. For reference, NHNM and NLNM represent the global high and low noise models, respectively. We have clarified this definition in the revised Fig. S5 caption and text in the supplementary file.**

---

## Author Comment (AC2)

General comments:

This paper presents receiver-function results from a deployment of OBS instruments in the Celebes Sea. The experiment encountered serious difficulties, with only nine stations usable for this analysis out of twenty-seven deployed. Add to this the difficulties of receiver-function analysis under typical OBS constraints (high noise level and short deployment time), and the authors have done a truly remarkable job getting the most out of this challenging data set. The main results, a Moho step near the Palu-Koro Fault and a low-velocity upper crust near the fault trace, are interesting and look to be robust (the Moho depths match very well between methods, for instance), and the authors sensibly don't place too much weight on less robust results, like the Vp/Vs values retrieved from H-k stacking. As noted below, the writeup is a bit brief and could stand to be clearer about some methodological aspects; I would consider this a minor revision.

(Note that I have deliberately not looked at other comments on this manuscript before writing this; it's best for reviews to be independent.)

Specific comments:

1. Some details on the OBS processing are missing. For instance, did the OBSes have hydrophones? If so, were they used to correct the seismic data in any way?

**Reply: Thank you for your comment. The preprocessing workflow of passive-source OBS data follows Yang et al. (2023) and has been described in detail in Text S1 and Text S2 of the supplementary file. The corresponding supplementary description has been added in the revised manuscript at Lines 90–100. We include the preprocessing steps our supplementary material, such as the azimuth correction and the time correction. Regarding the compliance noise correction using the hydrophone data, we did not do it since the hydrophone we used in this experiment are 20 – 100Hz , the low frequency response of the hydrophone is not low enough to remove the pressure noise from the vertical channels. As the low frequency hydrophone or pressure gauge gradually become a 'standard' OBS component nowadays. The hydrophone of the OBSs made by CAS are upgraded this year, so removing compliance noise becomes feasible in future.**

Clock drift correction: We observer up to several seconds per year from different OBS in our dataset and the cross-correlation approach provides a robust solution. We estimates and corrects clock errors in both land stations and OBS by leveraging ambient seismic noise cross-correlations across multiple components of motion follow the method by Hable et al. 2018 . Because noise-derived correlations should ideally be symmetric in causal vs. acausal parts, any systematic asymmetry is interpreted largely as a relative time offset (clock error) between the stations. The end result has errors of 0.5 ms with a linear drift assumption, and we applied the corrections linearly over the deployment period. Besides the quartz clock drift without GNSS, the OBS also employ several internal correction which are applied

automatically when exporting the data.

Azimuth correction: The azimuthal corrections of the horizontal components were performed following Niu and Li (2011). The coordinate system from the source to the OBS and the propagation (particle motion) direction of the P wave. A rotation step of 0.1° was applied to rotate the N–E coordinate system into the R–T coordinate system. We employed an SNR-weighted multi-event method to perform a grid search for the optimal back-azimuth angle $\theta_a$. At each grid point, the P-wave transverse component energy was calculated as $E_T(\theta_a)$.

2. For the receiver function deconvolution, what time-domain method was used? From context I would guess the iterdecon technique, but it's not stated explicitly.

Reply: Thank you for your comment. In this study, radial receiver functions were derived for the teleseismic P-wave arrivals using the time-domain iterative deconvolution technique of Ligorría and Ammon (1999). This information has been added in the revised manuscript at Lines 137–139.

3. In figure 3: what order are the RFs presented in? Chronological, or by epicentral distance, or something else? Also, it would be helpful to see an overlay of the expected arrival times for the best-fit H and k, to see what arrivals are being used.

Reply: Thank you for your comment. In Fig. 3, the receiver functions are presented in chronological order of the events. The H–κ stacking procedure automatically searches for the best-fit crustal thickness (H) and Vp/Vs ratio (κ), and therefore the specific phase arrivals used are not explicitly picked. As such, it is not straightforward to overlay individual predicted arrival times on each trace. Instead, the stacking process inherently accounts for the relevant converted phases and multiples to determine the optimal solution.

4. The text on page 7 refers to an anticorrelation between H and surface topography, but this isn't shown directly. A plot might make this more convincing (free-air gravity could also optionally be included). Figure 6 does show this, but doesn't include all stations.

Reply: Thank you for your comment. In the revised manuscript, the anticorrelation between topography and Moho depth is shown in Fig. 6c, where the red dashed line indicates the Moho depth beneath the seafloor topography in the upper panel. In addition, we have added a gravity anomaly profile in the supplementary file (Fig. S16) as supporting evidence for this result.

5. I assume that the traces were stacked before the RF waveform inversion, since a fit to only one trace is shown, but this isn't stated explicitly. Also, was a moveout applied before stacking? And in the inversion, were Vp and density held fixed? If so, where did the values come from?

**Reply: Thank you for your comment. In the OBS receiver function inversions, we used the Vp and density values from the nearby global model CRUST1.0 (Laske et al., 2013) and kept them fixed during the inversions, as receiver functions are not highly sensitive to these parameters. The same Vp values were also applied in the H–κ method to ensure consistency. In addition, the receiver function waveforms from all events at each station were linearly stacked prior to performing the waveform inversion. These clarifications have been added in the revised manuscript at Lines 196–199.**

6. In Figure 5, the H results look spatially coherent, but I don't think the k results do. The authors implicitly recognize this by not basing much interpretation on the k results, but it should be clearly stated in the text that a decision was made not to use them (this is not unusual for H-k results from noisy/limited data -- H is more robust).

**Reply: We thank the reviewer for pointing this out. Indeed, our Vp/Vs results are consistent with the low-velocity anomalies beneath the corresponding OBS stations. The apparent lack of spatial coherence mainly arises from the presence of thick sedimentary cover, which becomes progressively thicker toward the east. Because sediments are characterized by unusually high Vp/Vs ratios, they strongly affect the overall κ estimates. For example, at station C21F, where the sediment thickness is only about 0.2 km, the obtained κ value is very reasonable and close to the expected value for normal oceanic crust. In contrast, the sedimentary effects are more pronounced at stations located above the oceanic crust (C18F, C12F, C09G, C21F), the accretionary prism (C08F, C28F), and the thinned continental crust in the Makassar Strait (M01F, M02F, M03F). These sedimentary effects may introduce some uncertainties in the absolute κ values, but the relative variations among stations remain robust.**

[Figure]

**Figure R1: (a)** Thickness of the crust above the Moho; **(b)** Thickness of the oceanic crust; **(c)** Thickness of the sedimentary layer; **(d)** Vp/Vs ratios above the Moho; **(e)** Vp/Vs ratios of the oceanic crust; **(f)** Vp/Vs ratios of the sedimentary layer. (The black solid line represents the Palu–Koro fault (Patria and Putra, 2020), and the yellow solid circles represent the epicentre distribution of the earthquake after relocation.)

[Figure]

**Figure R2: Flowchart of the estimation procedure for sedimentary layer thickness and Vp/Vs ratio.**

7. The earthquake relocation work described on page 11 is a new result, so there should be a bit more detail on how it was obtained. What stations were used? Were the OBS data included?

Reply: Thank you for your comment. Since the Palu earthquake occurred in September 2018, while our OBS deployment took place from August 2019 to August 2020, no OBS data were available for this event. Therefore, we used the China Array network data as auxiliary information to assist in the relocation analysis. This result is only used as supporting information to assist in interpreting the tectonic setting, and this has been clarified in the revised manuscript at Lines 266–270.

**References**

Hable, S., Sigloch, K., Barruol, G., Stähler, S.C., and Hadziioannou, C.: Clock errors in land and ocean bottom seismograms: high-accuracy estimates from multiple-component noise cross-correlations, Geophysical Journal International., 214(3), 2014–2034, doi:10.1093/gji/ggy236, 2018.

Laske, G., Masters., G., Ma, Z. and Pasyanos, M.: Update on CRUST1.0–A 1–degree Global Model of Earth's Crust, Geophys. Res. Abstracts, 15, Abstract EGU2013–2658, 2013.

Ligorria, J.P., and Ammon, C.J.: Iterative deconvolution and receiver-function estimation, Bulletin of the Seismological Society of America., 89, 1395–1400, doi: 10.1785/BSSA0890051395, 1999.

Niu, F., and Li, J.: Component azimuths of the CEArray stations estimated from P-wave particle motion, Earthquake Science., 24, 3–13, doi:10.1007/s11589-011-0764-8, 2011.

Patria, A., and Putra, P.S.: Development of the Palu-Koro Fault in NW Palu Valley, Indonesia, Geoscience Letters., 7, doi:10.1186/s40562-020-0150-2, 2020.

Yang, T.W., Xu, Y., Nan, F.Z., Cao, D.P., Liu, L.H., and Hao, T.Y.: Construction and application of preprocessing technology for passive source ocean bottom seismometer data, Chinese Journal of Geophysics (in Chinese)., 66(2), 746–759, doi: 10.6038/cjg2022Q0076, 2023